# Nonlinear association between stress hyperglycemia ratio and severe consciousness disorder in acute ischemic stroke: A MIMIC retrospective analysis

**Zhangling Long[1], Shuang Liao[1], Ying Chen** [2]*

**1** Department of Neurology, Tongren City People's Hospital, Tongren, Guizhou, China, **2** Department of Neurology, The First People's Hospital of Guiyang, Guiyang, Guizhou, China

\* 532523783@qq.com

## Abstract

### Background

The stress hyperglycemia ratio (SHR) has been extensively studied; however, its association with severe consciousness disorder (Glasgow Coma Scale [GCS] ≤ 8) in patients with acute ischemic stroke (AIS) remains unclear. This study aimed to evaluate the association between SHR and GCS ≤ 8 in AIS as well as its relationship with long-term mortality.

### Methods

This retrospective cohort study based on the MIMIC database. The primary outcome was GCS ≤ 8, and the secondary outcome was long-term mortality. The Cox proportional risk model was used to evaluate the relationship between SHR and outcome, and the restricted cubic spline (RCS) method was used to explore the potential nonlinear relationship between SHR and outcome. In addition, Kaplan-Meier curves were used to assess the differences between SHR levels and the incidence of each outcome.

### Results

In this study, the overall incidence of GCS ≤ 8 and long-term mortality were 8.10% and 28.75%, respectively. Multivariate Cox regression analysis showed that SHR was associated with GCS ≤ 8 (HR = 1.52, 95%CI: 1.09–2.14, P = 0.015) and long-term mortality (HR = 1.32, 95%CI: 1.07–1.61, P < 0.0001), and RCS analysis showed a significant non-linear relationship between SHR and GCS ≤ 8 (P for non-linear <0.001), and an approximately linear relationship with long-term mortality (P for non-linear = 0.149). The Kaplan-Meier curve further confirmed that the incidence of GCS ≤ 8 and

**Data availability statement:** The data underlying the results of this study can be accessed from the MIMIC-IV database (https://physionet.org/content/mimiciv/2.2/). Since the data contain protected health information, access requires certification and completion of relevant training. Researchers who meet the access requirements can apply for access through the PhysioNet platform.

**Funding:** The author(s) received no specific funding for this work.

**Competing interests:** The authors have declared that no competing interests exist.

long-term mortality were significantly higher in patients with high SHR than in those with medium and low SHR (log-rank $P < 0.001$).

## Conclusions

Elevated SHR was associated with GCS ≤ 8 and long-term mortality in patients with AIS, with a nonlinear relationship for GCS ≤ 8. Further studies are required to confirm these results.

## 1. Introduction

Acute ischemic stroke (AIS) is a significant cause of mortality and long-term disability globally, accounting for a large proportion of stroke cases, and remains the second leading cause of death worldwide [1]. Despite significant advances in acute-phase treatments in recent years, the prognosis of patients with AIS remains poor, particularly in those with severe neurological dysfunction [2]. Therefore, identifying reliable prognostic biomarkers is crucial for risk stratification and early intervention in AIS patients.

Stress Hyperglycemia Ratio (SHR), a novel biomarker calculated as the ratio of blood glucose to glycated hemoglobin (HbA1c), reflects transient hyperglycemia during acute illness. This phenomenon is primarily driven by excessive secretion of catecholamines and cortisol during stress, leading to insulin resistance and increased gluconeogenesis [3,4]. Unlike chronic hyperglycemia, stress hyperglycemia represents a compensatory response to acute stress [5] and more accurately reflects an individual's baseline glycemic status and stress-induced hyperglycemia levels [6,7].

In recent years, SHR has garnered significant attention in the prognostic assessment of patients with AIS. Studies have shown that elevated SHR is significantly associated with adverse clinical outcomes in AIS patients, including early neurological deterioration (END), hemorrhagic transformation (HT), poor functional recovery, and both short- and long-term mortality [8–11]. AIS patients with consciousness disorders are known to have higher mortality rates and worse discharge outcomes [12]. However, the relationship between SHR, GCS ≤ 8, and prognosis in AIS patients remains unclear.

This study, based on the MIMIC database, aimed to clarify the association between SHR, GCS ≤ 8, and long-term mortality in patients with AIS. We hope to provide new theoretical evidence to support early identification and intervention of high-risk patients.

## 2. Method

### 2.1 Study population

This study used data from the Medical Information Mart for Intensive Care IV (MIMIC-IV, version 2.2) database. The MIMIC-IV is maintained by the Laboratory for Computational Physiology at the Massachusetts Institute of Technology and contains detailed clinical information on patients admitted to the Beth Israel Deaconess

Medical Center between 2008 and 2019. All patient data in the database were fully anonymized and contained no protected health information. Therefore, informed consent or ethical approval was not required. Ying Chen (ID: 62685292) obtained access to the database after completing training and certification.

The study population included patients diagnosed with cerebral infarction based on ICD-9 or ICD-10 codes. The exclusion criteria were as follows: (a) age < 18 years, (b) absence of blood glucose or hemoglobin test results at admission, (c) presence of GCS ≤ 8 prior to admission, and (d) extreme outliers in SHR values. Only the data from the first admission of patients with multiple hospitalizations were included in the analysis.

## 2.2 Patients

Patient-related data were extracted from the MIMIC-IV (version 2.2) database using Structured Query Language (SQL) via pgAdmin 4 (version 8.6). Demographic information included age, sex, race, marital status, and body mass index (BMI). Laboratory parameters included WBC count, lymphocyte count, neutrophil count, platelet count, hemoglobin, albumin, alanine aminotransferase (ALT), aspartate aminotransferase (AST), total bilirubin, blood urea nitrogen (BUN), serum creatinine, creatine kinase, creatine kinase-MB, blood glucose, glycated hemoglobin A1c (HbA1c), bicarbonate, chloride, potassium, total cholesterol (TC), and triglycerides. Comorbidities: identified using International Classification of Diseases, Ninth Revision(ICD-9) or Tenth Revision(ICD-10) codes, including hypertension, diabetes mellitus, hyperlipidemia, anemia, cancer, atrial fibrillation (AF), coronary artery disease (CAD), chronic kidney disease (CKD), respiratory failure (RF), heart failure (HF), and history of alcohol use or smoking. Medication use: Long-term use of antiplatelet agents or anticoagulants prior to the onset of a consciousness disorder.

The SHR was calculated using the following formula: SHR = (admission blood glucose (mg/dl))/(28.7 × HbA1c(%) − 46.7)] [13].

Handling of Missing Data: For variables with less than 20% missing data, multiple imputations based on a random forest model were used to estimate missing values. Continuous variables with more than 20% missing data were categorized based on the reference ranges provided by the database or by using the median or interquartile range and included in the analysis as dummy variables [14].

## 2.3 Outcome measures

The primary outcome was the occurrence of severe consciousness disorder during hospitalization, defined as a Glasgow Coma Scale (GCS) score ≤8. The secondary outcome was long-term mortality.

## 2.4 Statistical analysis

Continuous variables were expressed as mean ± standard deviation (SD) or median with interquartile range (IQR), while categorical variables were presented as counts and percentages. Differences in continuous variables were compared using Wilcoxon or Kruskal-Wallis tests, and differences in categorical variables were assessed using chi-squared tests. Multivariable Cox proportional hazards models were used to assess the association between SHR (both as a continuous variable and in tertiles) and outcomes of GCS ≤ 8 and long-term mortality. The results are presented as hazard ratios (HRs) with 95% confidence intervals (CIs). Adjustments for confounding variables were performed in three models: Model 1 was unadjusted. Model 2: Adjusted for age, sex, marital status, race, and BMI. Model 3: To avoid multicollinearity, variables with a variance inflation factor (VIF) ≥5 were excluded. This model included all variables from Model 2 plus albumin, ALT, BUN, creatine kinase, creatine kinase-MB, chloride, serum creatinine, hemoglobin, bicarbonate, potassium, lymphocyte count, platelet count, total cholesterol, triglyceride, (white blood cell count), AF, alcohol use, anemia, cancer, CAD, CKD, DM, HF, hypertension, hyperlipidemia, RF, tobacco use, anticoagulant drugs, and antiplatelet drugs. Moreover, RCS was utilized to examine the potential dose-response relationship between the SHR and outcomes, which was adjusted

for multiple covariates as previously mentioned [14]. Subgroup analyses were performed to investigate the association between the SHR and outcomes in different patient populations. A two-sided significance level of 0.05 was used for all analyses. Statistical analyses were performed with R software (version 4.4.1).

## 3. Results

### 3.1 Baseline characteristics

A total of 3,791 patients were included in this study (Fig 1), with a median age of 73 years (IQR61–83), and 49.93% were female. The overall GCS ≤ 8 was 8.10% (307/3,791) and the long-term mortality rate was 28.75% (1,090/3,791). Patients were divided into three groups based on their SHR values: Low (0.18–0.83), Medium (0.83–0.99), and High (0.99–3.30) (Table 1). The high SHR group exhibited significantly higher levels of blood urea nitrogen (BUN) and white blood cell counts compared to both the low SHR group and the medium SHR group (p < 0.001) (S3 Table). In terms of comorbidities, the prevalence of diabetes mellitus (DM) in the high SHR group was 40.82%, which was significantly greater than that in the low SHR group (p < 0.001) and the medium SHR group (p < 0.001). Additionally, the prevalence of heart failure (HF) was 21.84%, similarly being significantly higher than that in the low SHR group (p < 0.001) and the medium SHR group (p = 0.006)(S4 Table).

### 3.2 Cox proportional hazard analysis

Cox regression analysis showed a significant association between SHR, GCS ≤ 8, and long-term mortality. In the unadjusted model (model 1), SHR was significantly positively associated with GCS ≤ 8 (HR = 2.41, 95% CI: 1.83–3.17, P < 0.0001) and long-term mortality (HR = 1.90, 95% CI: 1.59–2.26, P < 0.0001).After adjusting for demographic characteristics (model 2), the associations between SHR and GCS ≤ 8 (HR = 2.33, 95% CI: 1.73–3.12, P < 0.0001) and long-term mortality (HR = 1.97, 95% CI: 1.64–2.36, P < 0.0001) remained significant. Further adjustment for clinical variables (model 3)

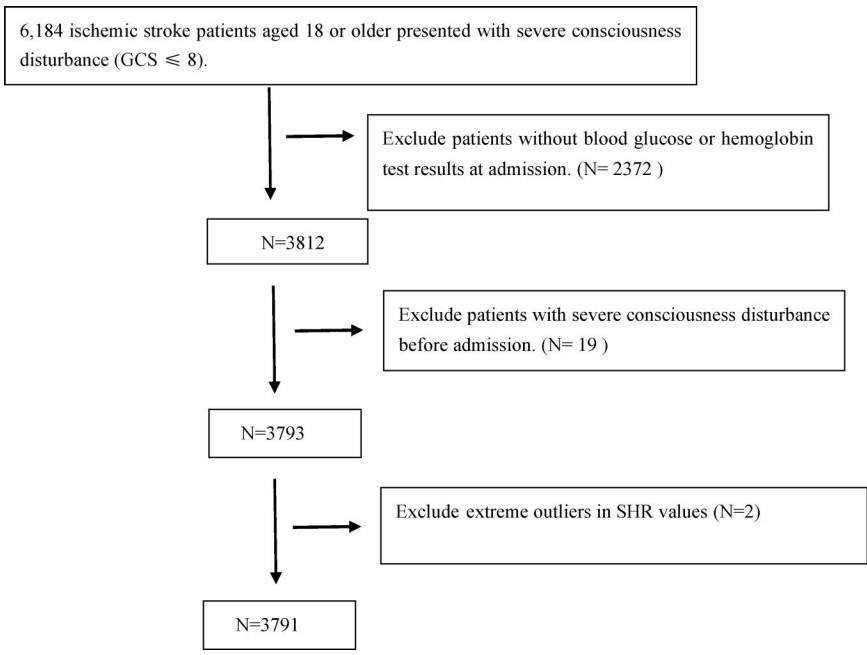

**Fig 1. Flow chart of patient selection.**

**Table 1. Baseline characteristics by SHR tertiles (N = 3791).**

| Parameters | Low | Medium | High | P-value |
|---|---|---|---|---|
| | N = 1264 | N = 1263 | N = 1264 | |
| Age, years | 74 [63-84] | 72 [61-82] | 73 [61-83] | 0.015 |
| BUN,mg/dl | 17.00 (13.00-22.00) | 16.00 (12.00-21.00) | 18.00(13.00-24.00) | <0.001 |
| Chloride,mEq/L | 104.02 (4.15) | 103.72 (3.96) | 102.89 (4.40) | <0.001 |
| Serum creatinine,mg/dL | 0.90 (0.80-1.20) | 0.90 (0.70-1.10) | 0.90 (0.80-1.20) | <0.001 |
| blood glucose,mg/dL | 99.75 (28.57) | 114.91 (32.52) | 166.90 (71.41) | <0.001 |
| hemoglobin,g/dL | 12.37 (1.88) | 12.69 (1.92) | 12.34 (2.11) | <0.001 |
| HBA1C% | 6.57 (1.63) | 6.07 (1.23) | 6.28 (1.44) | <0.001 |
| bicarbonate,mEq/L | 25.28 (3.45) | 24.99 (2.97) | 24.14 (3.44) | <0.001 |
| Potassium,mEq/L | 4.03 (0.47) | 4.04 (0.50) | 4.09 (0.58) | 0.004 |
| Platelet,k/uL | 233.95 (83.39) | 228.40 (86.29) | 223.79 (89.85) | 0.013 |
| total cholesterol,mg/dL | 167.74 (44.22) | 173.15 (43.73) | 167.23 (49.63) | 0.002 |
| Triglyceride,mg/dL | 110.00 (80.00-146.00) | 108.00 (81.00-147.00) | 114.05 (83.00-154.00) | 0.026 |
| white blood cell, k/uL | 7.40 (6.00-9.20) | 7.80 (6.30-9.90) | 9.00 (6.90-11.90) | <0.001 |
| Gender | | | | 0.086 |
| Male | 604 (47.78%) | 659 (52.18%) | 635 (50.24%) | |
| Female | 660 (52.22%) | 604 (47.82%) | 629 (49.76%) | |
| BMI | | | | 0.067 |
| <23.9 | 158 (12.50%) | 131 (10.37%) | 136 (10.76%) | |
| 23.9-27.9 | 205 (16.22%) | 174 (13.78%) | 177 (14.00%) | |
| >27.9 | 271 (21.44%) | 296 (23.44%) | 323 (25.55%) | |
| Missing | 630 (49.84%) | 662 (52.41%) | 628 (49.68%) | |
| Marital status | | | | <0.001 |
| Married | 588 (46.52%) | 598 (47.35%) | 557 (44.07%) | |
| Single | 270 (21.36%) | 279 (22.09%) | 298 (23.58%) | |
| Widowed | 248 (19.62%) | 212 (16.79%) | 212 (16.77%) | |
| Divorced | 101 (7.99%) | 88 (6.97%) | 61 (4.83%) | |
| Missing | 57 (4.51%) | 86 (6.81%) | 136 (10.76%) | |
| Race | | | | <0.001 |
| White | 840 (66.46%) | 894 (70.78%) | 833 (65.90%) | |
| Other | 164 (12.97%) | 189 (14.96%) | 247 (19.54%) | |
| Black | 222 (17.56%) | 145 (11.48%) | 157 (12.42%) | |
| Asian | 38 (3.01%) | 35 (2.77%) | 27 (2.14%) | |
| albumin,g/dl | | | | <0.001 |
| <3.8 | 261 (20.65%) | 207 (16.39%) | 292 (23.10%) | |
| >=3.8 | 247 (19.54%) | 325 (25.73%) | 298 (23.58%) | |
| Missing | 756 (59.81%) | 731 (57.88%) | 674 (53.32%) | |
| ALT,iu/l | | | | 0.004 |
| <18 | 360 (28.48%) | 351 (27.79%) | 327 (25.87%) | |
| >=18 | 352 (27.85%) | 386 (30.56%) | 442 (34.97%) | |
| Missing | 552 (43.67%) | 526 (41.65%) | 495 (39.16%) | |
| AST,iu/l | | | | <0.001 |
| <22 | 370 (29.27%) | 342 (27.08%) | 306 (24.21%) | |
| >=22 | 348 (27.53%) | 397 (31.43%) | 468 (37.03%) | |
| Missing | 546 (43.20%) | 524 (41.49%) | 490 (38.77%) | |

*(Continued)*

Table 1. (Continued)

| Parameters | Low | Medium | High | P-value |
|---|---|---|---|---|
| | N = 1264 | N = 1263 | N = 1264 | |
| creatine kinase,iu/l | | | | <0.001 |
| 47-322 | 452 (35.76%) | 494 (39.11%) | 490 (38.77%) | |
| <47 | 115 (9.10%) | 92 (7.28%) | 102 (8.07%) | |
| >322 | 63 (4.98%) | 88 (6.97%) | 116 (9.18%) | |
| Missing | 634 (50.16%) | 589 (46.63%) | 556 (43.99%) | |
| creatine kinase-MB,ng/mL | | | | <0.001 |
| <3 | 271 (21.44%) | 237 (18.79%) | 233 (18.43%) | |
| 3.0-10 | 317 (25.08%) | 354 (28.07%) | 339 (26.82%) | |
| >10 | 19 (1.50%) | 36 (2.85%) | 75 (5.93%) | |
| Missing | 657 (51.98%) | 634 (50.28%) | 617 (48.81%) | |
| Lymphocyte, k/uL | | | | <0.001 |
| <1.08 | 39 (3.09%) | 55 (4.35%) | 95 (7.52%) | |
| 1.08-1.74 | 58 (4.59%) | 57 (4.51%) | 85 (6.72%) | |
| >1.74 | 62 (4.91%) | 51 (4.04%) | 75 (5.93%) | |
| Missing | 1105 (87.42%) | 1100 (87.09%) | 1009 (79.83%) | |
| Neutrophil,k/uL | | | | <0.001 |
| <5.1 | 64 (5.06%) | 59 (4.67%) | 66 (5.22%) | |
| 5.1-8.1 | 65 (5.14%) | 60 (4.75%) | 72 (5.70%) | |
| >8.1 | 30 (2.37%) | 44 (3.48%) | 117 (9.26%) | |
| Missing | 1105 (87.42%) | 1100 (87.09%) | 1009 (79.83%) | |
| total bilirubin,mg/dL | | | | <0.001 |
| <0.4 | 164 (12.97%) | 133 (10.53%) | 150 (11.87%) | |
| 0.4-0.7 | 371 (29.35%) | 360 (28.50%) | 372 (29.43%) | |
| >0.7 | 121 (9.57%) | 188 (14.89%) | 198 (15.66%) | |
| Missing | 608 (48.10%) | 582 (46.08%) | 544 (43.04%) | |
| AF | 265 (20.97%) | 258 (20.43%) | 297 (23.50%) | 0.135 |
| Alcohol use | 13 (1.03%) | 22 (1.74%) | 12 (0.95%) | 0.14 |
| Anemia | 151 (11.95%) | 135 (10.69%) | 255 (20.17%) | <0.001 |
| Cancer | 90 (7.12%) | 92 (7.28%) | 101 (7.99%) | 0.676 |
| CAD | 339 (26.82%) | 295 (23.36%) | 382 (30.22%) | <0.001 |
| CKD | 103 (8.15%) | 95 (7.52%) | 105 (8.31%) | 0.744 |
| diabetes mellitus | 412 (32.59%) | 329 (26.05%) | 516 (40.82%) | <0.001 |
| HF | 220 (17.41%) | 163 (12.91%) | 276 (21.84%) | <0.001 |
| hypertension | 495 (39.16%) | 501 (39.67%) | 519 (41.06%) | 0.601 |
| hyperlipidemia | 495 (39.16%) | 481 (38.08%) | 405 (32.04%) | <0.001 |
| RF | 64 (5.06%) | 73 (5.78%) | 166 (13.13%) | <0.001 |
| Tobacco use | 99 (7.83%) | 102 (8.08%) | 75 (5.93%) | 0.076 |

Data are expressed as mean(SD),median (Q1-Q3) or N(%) Abbreviations: BMI (body mass index), ALT(alanine aminotransferase), AST(aspartate aminotransferase), HbA1c (hemoglobin a1c), AF(Atrial Fibrillation), CHD(coronary heart disease), CKD(chronic kidney disease), HF(heart failure), RF(respiratory failure), SHR(stress hyperglycemia ratio)

SHR tertiles: Low (0.18–0.83), Medium (0.83–0.99), and High (0.99–3.30)

showed that SHR was still significantly associated with GCS ≤ 8 (HR = 1.52, 95% CI: 1.09–2.14, P = 0.015) and long-term mortality (HR = 1.32, 95% CI: 1.07–1.61, P < 0.0001).SHR tertiles showed that the high group (0.99–3.30) had a significantly increased risk of GCS ≤ 8(HR = 1.78, 95% CI: 1.30–2.44, P = 0.0003) and long-term mortality (HR = 1.44, 95% CI: 1.24–1.68, P < 0.0001) compared with the low group (0.18–0.83). Trend tests also showed a significant positive correlation between increasing SHR and GCS ≤ 8 (P < 0.0001) and long-term mortality (P < 0.0001) (Table 2).

### 3.3 Kaplan–Meier survival curve analysis

Kaplan-Meier survival curves showed that after grouping by SHR tertile, the cumulative incidence of GCS ≤ 8 was significantly higher in the High SHR group than in the Medium and Low SHR groups (log-rank test, P < 0.0001). Similarly, the long-term survival mortality in the High SHR group was significantly lower than that in the Medium and Low SHR groups (log-rank test, P < 0.0001) (Fig 2).

### 3.4 Non-linear association between SHR and outcomes

Restricted cubic spline analysis was performed to explore the potential nonlinear relationship between the SHR and outcomes. The results showed a significant nonlinear association between increasing SHR and GCS ≤ 8 (P for nonlinearity < 0.001). However, after full adjustment for confounders, the relationship between SHR and long-term mortality was approximately linear (P for nonlinearity = 0.149(Fig 3).

### 3.5 Subgroup analyses

Subgroup analysis showed that SHR was significantly associated with GCS ≤ 8 in most subgroups, except in patients with cancer (P = 0.218), with no significant interactions observed between subgroups (Fig 4a).

**Table 2. Cox proportional hazard ratios.**

| outcomes | Model 1 | P value | Model 2 | P value | Model 3 | P value |
|---|---|---|---|---|---|---|
| | HR (95% CI) | | HR (95% CI) | | HR (95% CI) | |
| **gcs ≤ 8** | | | | | | |
| SHR | 2.41 (1.83, 3.17) | <0.0001 | 2.33 (1.73, 3.12) | <0.0001 | 1.52 (1.09, 2.14) | 0.015 |
| SHR Tertile[a] | | | | | | |
| Low, N = 1264 | Ref | | Ref | | Ref | |
| Medium, N = 1263 | 1.19 (0.84, 1.67) | 0.3336 | 1.11 (0.79, 1.57) | 0.5427 | 1.09 (0.76, 1.55) | 0.642 |
| High, N = 1264 | 2.41 (1.79, 3.24) | <0.0001 | 2.14 (1.58, 2.89) | <0.0001 | 1.78 (1.30, 2.44) | 0.0003 |
| P for trend | | <0.0001 | | <0.0001 | | <0.0001 |
| **long-term mortality** | | | | | | |
| SHR | 1.90 (1.59, 2.26) | <0.0001 | 1.97 (1.64, 2.36) | <0.0001 | 1.32 (1.07, 1.61) | <0.0001 |
| SHR Tertile[a] | | | | | | |
| Low, N = 1264 | Ref | | Ref | | Ref | |
| Medium, N = 1263 | 0.99 (0.85, 1.16) | 0.8834 | 0.97 (0.83, 1.13) | 0.6681 | 0.98 (0.83, 1.15) | 0.8074 |
| High, N = 1264 | 1.66 (1.44, 1.91) | <0.0001 | 1.73 (1.50, 2.00) | <0.0001 | 1.44 (1.24, 1.68) | <0.0001 |
| P for trend | | <0.0001 | | <0.0001 | | <0.0001 |

Model 1 was unadjusted.

Model 2 was adjusted for: Age, Gender; Marital status; Race, BMI

Model 3 was adjusted Model2 and albumin, ALT, BUN, creatine kinase, creatine kinase-MB, Chloride, serum creatinine, hemoglobin, bicarbonate, Potassium, Lymphocyte, Platelet, total cholesterol, Triglyceride, white blood cell, AF, Alcohol use, Anemia, Cancer, CHD, CKD, diabetes mellitus, HF, hypertension, hyperlipidemia, RF, Tobacco use, Anticoagulant drugs, Antiplatelet drugs. HR (Hazard Ratio), CI (Confidence Interval)

SHR Tertile[a]: Low:0.18–0.83; Medium: 0.83–0.99; High: 0.99–3.30

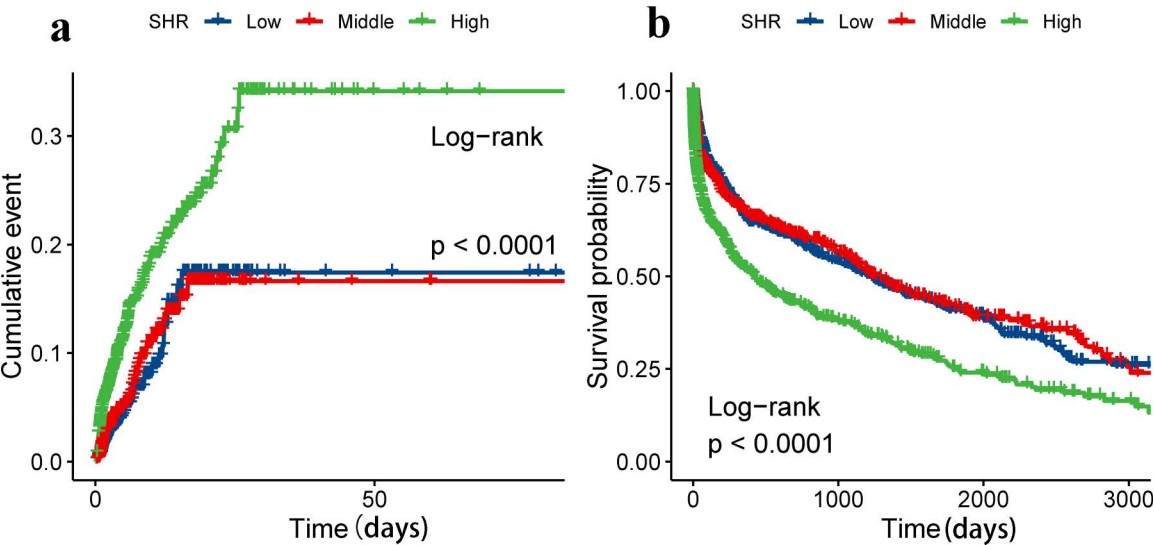

**Fig 2. Cumulative event and survival incidence curves.** (SHR Tertile: Low 0.18-0.83; Medium 0.83-0.99; High 0.99-3.30). a: Cumulative event incidence curves for incidence of GCS ≤ 8. b: Survival curves for long-term mortality in the entire study population.

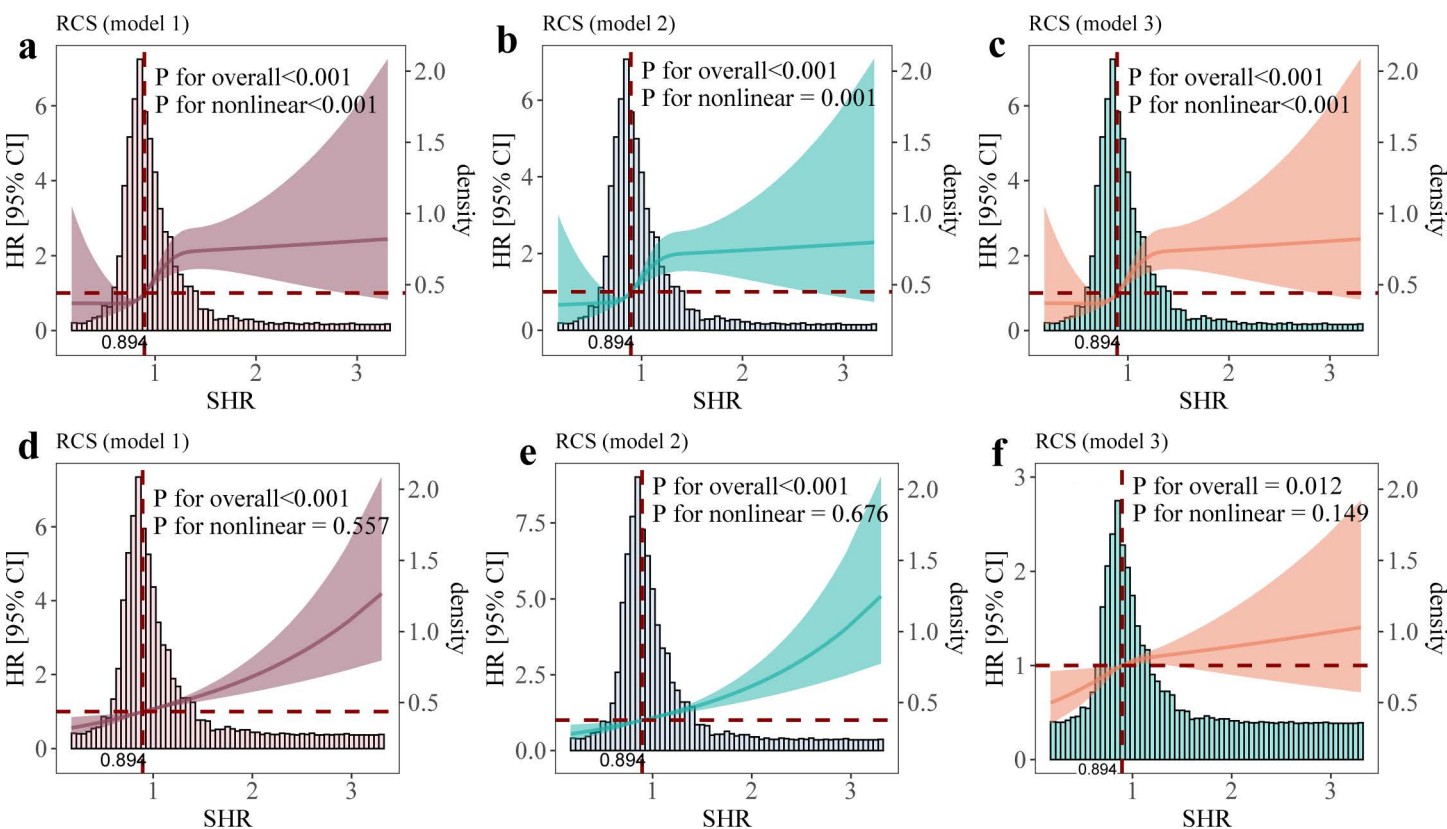

**Fig 3. RCS curves for the HR and distribution of SHR. (a)**, **(b)**, **(c)**: GCS ≤ 8 cumulative incidence curves for Model 1, Model 2, and Model 3. **(d)**, **(e)**, **(f)**: Long-term mortality survival curves and histograms for Model 1, Model 2, and Model 3. Model 1 was unadjusted. Model 2 was adjusted for gender, age, race, and BMI. Model 3 was adjusted for the variables in model 2 and further adjusted for albumin, ALT, BUN, creatine kinase, creatine kinase-MB, Chloride, serum creatinine, hemoglobin, bicarbonate, Potassium, Lymphocyte, Platelet, total cholesterol, Triglyceride, white blood cell, AF, Alcohol use, Anemia, Cancer, CHD, CKD, diabetes mellitus, HF, hypertension, hyperlipidemia, RF, Tobacco use, Anticoagulant drugs, Antiplatelet drugs.

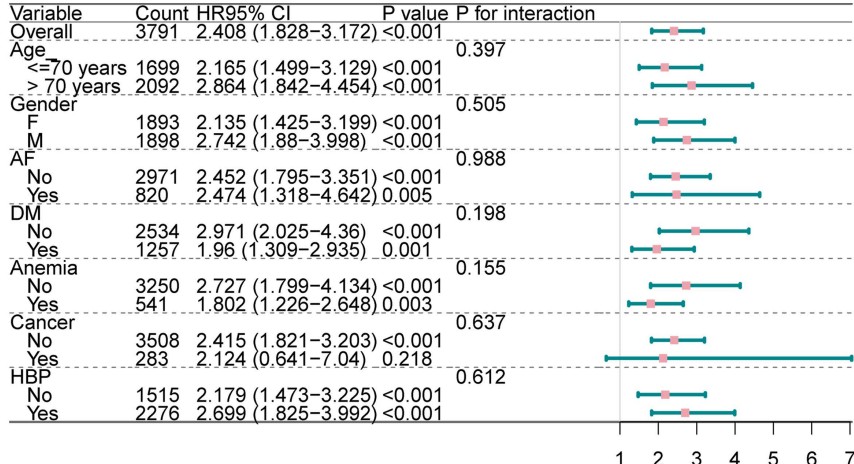

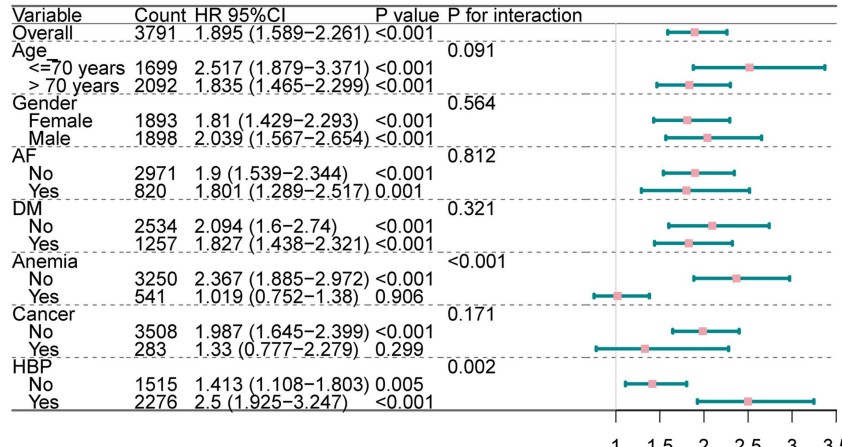

**Fig 4. Forest plots of stratified analyses of SHR and outcomes.** a: Forest plot of HRs for GCS ≤ 8 in different subgroups. b: Forest plots of HRs for long-term mortality in different subgroups. DM (diabetes mellitus), AF (Atrial Fibrillation), HBP(high blood pressure).

For long-term mortality, SHR was significantly associated in most subgroups, except in patients with anemia (P = 0.906) and cancer (P = 0.299). Significant interactions were found in the anemia (P < 0.001) and high blood pressure subgroups (P = 0.002) (Fig 4b).

## Discussion

This study, based on the MIMIC-IV database of patients with stroke, used a retrospective cohort design to evaluate the association between the stress hyperglycemia ratio (SHR) and GCS ≤ 8 and long-term mortality in patients with AIS. To the best of our knowledge, this is the first study to identify a significant nonlinear association between elevated SHR and GCS ≤ 8. Furthermore, the relationship between SHR and long-term mortality was found to follow an approximately linear trend.

As a novel biomarker, SHR has demonstrated unique clinical advantages in cardiovascular disease [15,16], intensive care [17], acute illness [18] and diabetes management [19]. It has also shown significant clinical value in the prognostic evaluation of stroke patients. Studies have reported that increased SHR is associated with hematoma volume in

intracerebral hemorrhage, 48-hour and 30-day mortality, and poor 3-month outcome (modified Rankin Scale [mRS] 4–6) [20]. In patients with AIS, SHR has been independently associated with moderate-to-severe cerebral edema, poor functional outcome, hemorrhagic transformation, in-hospital mortality, and prolonged hospital stay [8,21–23]. Another study of 1,376 critically ill AIS patients admitted to the ICU found that SHR was associated with increased 30-day, 90-day, and 1-year mortality, regardless of diabetes status [24]. Furthermore, a meta-analysis of 183,588 patients found that higher SHR significantly increased the risk of adverse outcomes, mortality, neurological deficits, hemorrhagic transformation and infectious complications, regardless of diabetes status or the use of intravenous thrombolysis or mechanical thrombectomy [25].

GCS ≤ 8 in patients AIS is a critical marker of disease severity and is associated with a high incidence of complications and comorbidities, as well as increased in-hospital mortality, 3-month mortality, and disability rates [26,27]. Furthermore, impaired consciousness at discharge has been linked to significantly worse long-term outcomes following AIS [28]. In this study, we found that the incidence of GCS ≤ 8 in patients with AIS was 8.10%, and the long-term mortality rate was 28.75%. Patients with GCS ≤ 8 had significantly higher mortality than those with GCS > 8 (59.28% vs. 26.06%, P < 0.001), and their SHR values were also higher (1.03 [0.86–1.19] vs. 0.89 [0.78–1.04]) (S1 Table). These findings highlight the strong association between GCS ≤ 8 and poor outcomes in AIS patients. Our analysis of SHR tertiles revealed an approximately linear relationship between elevated SHR and long-term mortality (P for nonlinear > 0.05), suggesting that SHR may serve as a reliable predictor of long-term mortality in AIS patients. However, the relationship between SHR and GCS ≤ 8 was nonlinear (P for nonlinear < 0.001), indicating that the impact of SHR on GCS ≤ 8 may involve more complex mechanisms. These findings suggest that SHR not only reflect the severity of stress hyperglycemia but may also capture other pathophysiological processes contributing to poor outcomes in AIS patients.

The potential mechanisms by which elevated SHR contributes to GCS ≤ 8 and increased long-term mortality in AIS patients may involve several pathways. Hyperglycemia has been shown to activate the NF-κB signaling pathway and promotes the release of pro-inflammatory cytokines, leading to a systemic inflammatory response and exacerbating brain tissue injury [29,30]. In our study, we observed that in the SHR tertile analysis, the high group had significantly higher white blood cell counts and a greater proportion of neutrophils >8.1 k/uL compared to the low group (P < 0.001), further supporting the role of inflammation in poor outcomes. Additionally, hyperglycemia increases the production of reactive oxygen species (ROS), which induces oxidative stress and cellular damage. This process can impair the integrity of cerebrovascular endothelial cells, increase blood-brain barrier permeability, and exacerbate cerebral edema and neuronal injury [3]. Moreover, hyperglycemia can worsen insulin resistance, disrupt normal glucose metabolism, and lead to an insufficient energy supply to brain cells. This energy deficit may impair neuronal function and survival, further aggravating neurological damage, and contributing to poor outcomes [31]. These effects may contribute to the development of GCS ≤ 8 and poor functional outcomes in AIS patients.

This study had several limitations. First, it was conducted using a retrospective design, which may introduce selection bias and unadjusted confounders that may affect the interpretation of the results. Second, the study was based on the MIMIC database, which has a limited scope and does not include all variables that may influence the relationship between SHR and patient outcomes, such as lifestyle factors, Albumin-Corrected Anion Gap [32], socioeconomic status, and genetic predisposition. However, we used the E-value sensitivity analysis to quantify the potential implications of unmeasured confounders and found that an unmeasured confounder was unlikely to explain the entirety of the effect [33]. Third, the MIMIC database consists primarily of data from the United States, with a predominance of white populations, which limits the generalizability of the findings to other racial or ethnic groups. Future prospective multicenter studies with long-term follow-up are needed to validate these findings and to address the above limitations. Such studies will provide a more comprehensive assessment of the prognostic value of SHR in AIS patients and provide more robust evidence for risk stratification and management in different populations.

## Conclusion

This study, based on 3,791 patients with acute ischemic stroke (AIS) from the MIMIC database, found that SHR was significantly associated with GCS ≤ 8 and long-term mortality. A nonlinear relationship was observed between SHR and GCS ≤ 8, suggesting that SHR could serve as an important indicator for risk stratification and precise management in patients with AIS.

## Supporting information

**S1 Table. Baseline characteristics of the GCS > 8 and GCS ≤ 8 groups.** Data are expressed as mean (SD), median (Q1-Q3) or N (%). Abbreviations: BMI (body mass index), ALT(alanine aminotransferase), AST(aspartate aminotransferase), BUN (blood urea nitrogen), HbA1c (hemoglobin a1c), AF(Atrial Fibrillation), CHD(coronary heart disease), CKD(chronic kidney disease), HF(heart failure), RF(respiratory failure), SHR(stress hyperglycemia ratio)
(DOCX)

**S2 Table. Shapiro-wilk test.**
(DOCX)

**S3 Table. Statistical analysis of continuous variables using dunn's test with bonferroni correction across SHR tertiles.**
(DOCX)

**S4 Table. Statistical evaluation of categorical variables by shr tertiles using chi-square and fisher's exact tests with FDR correction.**
(DOCX)

## Author contributions

**Conceptualization:** Zhangling Long.

**Data curation:** Ying Chen.

**Formal analysis:** Zhangling Long.

**Investigation:** Shuang Liao.

**Methodology:** Ying Chen.

**Project administration:** Ying Chen.

**Resources:** Zhangling Long.

**Software:** Ying Chen.

**Supervision:** Shuang Liao.

**Validation:** Shuang Liao.

**Visualization:** Shuang Liao.

**Writing – original draft:** Ying Chen, Zhangling Long.

**Writing – review & editing:** Ying Chen, Zhangling Long.

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
