## [Decision Letter · Decision Letter 0]

16 Apr 2025

Dear Dr. Chen,

Thank you for submitting your manuscript to PLOS ONE. After careful consideration, we feel that it has merit but does not fully meet PLOS ONE’s publication criteria as it currently stands. Therefore, we invite you to submit a revised version of the manuscript that addresses the points raised during the review process.

Provide clear justification for further revision: The major concern with this paper is that it represents data (except SHR as the independent variable) similar to recently published article by the *Corresponding author* , “Association of albumin-corrected anion gap with severe consciousness disorders and outcomes in ischemic stroke: a retrospective MIMIC analysis”, https://doi.org/10.1038/s41598-024-76324-x. The two papers appear to be parts of same exploratory analysis, not different hypothesis-driven studies.It is recommended to mention that weather the same datasets were used for the present study or different.It is necessory to clearly explain in the methodology section regarding the need of three different statistical test used to analyse the data in the present study.Provide the major concerns regarding the data raised by reviewer 2.

We look forward to receiving your revised manuscript.

Kind regards,

Tapan Amrutlal Patel, Ph.D.

Academic Editor

PLOS ONE

Journal Requirements:

1. When submitting your revision, we need you to address these additional requirements.c

Reviewers' comments:

Reviewer's Responses to Questions

**Comments to the Author**

1. Is the manuscript technically sound, and do the data support the conclusions?

Reviewer #1: Yes

Reviewer #2: Partly

2. Has the statistical analysis been performed appropriately and rigorously?

Reviewer #1: Yes

Reviewer #2: No

3. Have the authors made all data underlying the findings in their manuscript fully available?

Reviewer #1: No

Reviewer #2: No

4. Is the manuscript presented in an intelligible fashion and written in standard English?

Reviewer #1: Yes

Reviewer #2: Yes

Reviewer #1: The topic discussed is very interesting and innovative for the management and prognosis of patients with stroke. Although I must make some small suggestions.

General:

1. You should not use too many abbreviations or acronyms, even less so if the term is only one word. This makes it difficult to read

2. The data underlying the findings must be made available without restrictions. Please specify where they can be accessed, or explain the reasons why they cannot be shared.

3. In section 2.2 Patients, the authors repeat a sentence.

4. In figure 2, specify the time unit.

Reviewer #2: 1. My main concern with this paper is that it is similar to one previously published by the first author, “Association of albumin-corrected anion gap with severe consciousness disorders and outcomes in ischemic stroke: a retrospective MIMIC analysis”, https://doi.org/10.1038/s41598-024-76324-x. The data, statistical tests, and figures are all the same, the only substantive difference being that this new paper uses SHR as the independent variable while the previous paper treats ACAG as the independent variable. The two papers appear to be parts of a single exploratory analysis, not distinct hypothesis-driven studies.

2. The reasons for running three different series of statistical tests (ANOVA/chi-squared tests on group means, Cox hazard models to test for low GCS event likelihood conditional on SHR, and a nonlinear RCS regression model) were not given. It seems that the Cox models are the most theoretically appropriate analysis to run, in which case, the group-mean tests are superfluous. The Cox models are already nonlinear (with multiplicative exponential effects), so it’s not clear why another type of nonlinear test (RCS) should be run, or how to interpret the results of the two models in relation to each other.

3. The initial group mean tests (ANOVA/chi-squared, table 1) include not only the target dependent variable GCS, but the independent variable SHR (redundant) and many other variables which the paper refers to as “confounding variables” (line 141). If these are merely confounding variables which need to be controlled for in a statistical model, then putting them into the model as additional independent variables suffices. There is no point to running statistical tests on whether these variables themselves have means which vary with the primary independent variable of interest.

4. There is no discussion of any multiple-comparisons corrections, but such corrections seem to be needed. I counted 104 tests reported in the paper, many with a p-value evidently between 0.001 and 0.0001, which means many of those p-values are not likely to remain above a significance threshold of 0.05 after correction (because 0.001 x 100 = 0.1). In addition, the fact that the analysis here so closely parallels the one published in the ACAG paper raises the worry that the actual number of tests run on the data is higher than reported here.

5. The paper is not clear in many areas. For example, line 131 is a description of table 1 that should go in a caption, not part of the statistical analysis. This paragraph makes no reference to table 1, and I did not understand it until I saw table 1. It would be helpful if the author’s published the R code they used, as someone could only reconstruct the nuts-and-bolts of their analysis with considerable effort.

6. Some of the references have issues. For example, the first line of the paper says that AIS “is the leading cause of death and long-term disability worldwide”, but this is not true, and the reference given contradicts it.

**Do you want your identity to be public for this peer review?** For information about this choice, including consent withdrawal, please see our Privacy Policy

Reviewer #1: **Yes: ** Leonardo Albitres-Flores

Reviewer #2: No

---

## [Author Response · Author response to Decision Letter 1]

10 May 2025

Thank you for your thorough review and valuable feedback on my manuscript titled “[Nonlinear association between stress hyperglycemia ratio and severe consciousness disorder in acute ischemic stroke: A MIMIC retrospective analysis]” (Manuscript ID: PONE-D-24-59138). I have taken all comments seriously and revised the manuscript according to the reviewers' suggestions.

---

## [Decision Letter · Decision Letter 1]

21 Jul 2025

Nonlinear association between stress hyperglycemia ratio and severe consciousness disorder in acute ischemic stroke: A MIMIC retrospective analysis

PONE-D-24-59138R1

Dear Dr. Chen,

We’re pleased to inform you that your manuscript has been judged scientifically suitable for publication and will be formally accepted for publication once it meets all outstanding technical requirements.

Kind regards,

Tapan Amrutlal Patel, Ph.D.

Academic Editor

PLOS ONE

Additional Editor Comments (optional):

Reviewers' comments:

Reviewer's Responses to Questions

**Comments to the Author**

Reviewer #1: All comments have been addressed

2. Is the manuscript technically sound, and do the data support the conclusions?

Reviewer #1: Yes

3. Has the statistical analysis been performed appropriately and rigorously?

Reviewer #1: Yes

4. Have the authors made all data underlying the findings in their manuscript fully available?

Reviewer #1: Yes

5. Is the manuscript presented in an intelligible fashion and written in standard English?

Reviewer #1: Yes

Reviewer #1: (No Response)

**Do you want your identity to be public for this peer review?** For information about this choice, including consent withdrawal, please see our Privacy Policy

Reviewer #1: No

---

## [Editor Report · Acceptance letter]

PONE-D-24-59138R1

PLOS ONE

Dear Dr. Chen,

I'm pleased to inform you that your manuscript has been deemed suitable for publication in PLOS ONE. Congratulations! Your manuscript is now being handed over to our production team.

Kind regards,

on behalf of

Dr. Tapan Amrutlal Patel

Academic Editor

PLOS ONE